# The Effect of Losartan on Neuroinflammation as Well as on Endothelin-1- and Serotonin-Induced Vasoconstriction in a Double-Haemorrhage Rat Model

**DOI:** 10.3390/jcm11247367

**Published:** 2022-12-12

**Authors:** Jürgen Konczalla, Jan Mrosek, Sepide Kashefiolasl, Christian Musahl, Serge Marbacher, Gerrit Alexander Schubert, Lukas Andereggen, Stefan Wanderer

**Affiliations:** 1Department of Neurosurgery, Goethe-University Hospital, Schleusenweg 2-16, 60528 Frankfurt am Main, Germany; 2Department of Neurosurgery, Kantonsspital Aarau, Tellstrasse 25, 5001 Aarau, Switzerland

**Keywords:** neuroinflammation, aneurysm, subarachnoid haemorrhage, losartan, neurorestauration

## Abstract

Poor patient outcome after aneurysmal subarachnoid haemorrhage (SAH) is due to a multifactorial process. Delayed cerebral vasospasm, ischemic neurological deficits, and infarction are the most feared acute sequelae triggered by enhanced synthesis of serotonin and endothelin-1 (ET-1). During the past decades, multiple drugs have been analysed for protective effects without resounding success. Therefore, the authors wanted to analyse the potential beneficial role of Losartan (LOS). Male Sprague Dawley rats were randomised into either a group receiving two injections of blood into the cisterna magna (SAH group) or a group receiving two injections of isotonic sodium chloride (sham group). The animals were culled on day five and basilar artery ring segments were used for in vitro tension studies. Sarafotoxin S6c caused a dose-dependent vasorelaxation in sham and SAH segments, which was more pronounced in sham segments. LOS, applied in a concentration of 10^−3^ M, was able to significantly reduce serotonin- (*p* < 0.01) and ET-1- (*p* < 0.05, *p* < 0.01) mediated vasoconstriction in sham segments. These findings, along with the well-known beneficial effects of LOS on restoring the impaired endothelin-B_1_-receptor function after SAH, as well as on the neuroprotectional and antiepileptogenic aspects, might be implemented in advancing tailored concepts to sufficiently ameliorate patients’ functional outcome after SAH.

## 1. Introduction

To date, subarachnoid haemorrhage (SAH) due to aneurysm rupture remains a potentially devastating neurosurgical condition, possibly leading to severe disability, dependency, or death [1]. Despite intensive research efforts in minimising SAH-associated complications, such as microcirculatory disturbances, cortical spreading depressions, hydrocephalus, and early intracranial pressure elevation, patients’ overall outcomes are mostly poor [2]. It is already well-known that proinflammatory and vasoconstrictive cytokines and peptides, such as interleukins, serotonin (5-HT), and endothelin-1 (ET-1), are key players in elucidating delayed cerebral vasospasm (DCVS) in macro- and microcirculatory aspects [3,4]. ET-1 mediates DCVS via a vasoconstrictory endothelin-A-receptor (ET-A-R), located on the muscular vessel media, whereas vasorelaxation is physiologically mediated via the endothelin-B1-receptor (ET-B_1_-R) of the endothelium [5]. Under pathophysiological circumstances like SAH, the ET-B_1_-R function is impaired, while the ET-1 synthesis is enhanced, and therefore the vasocontractile response of the ET-A-R is pronounced as well [6,7]. Besides inflammatory cytokines and peripheral immune cells, prostaglandin F2alpha is additively involved in maintaining cerebral inflammation [8,9].

Losartan (LOS), an angiotensin-2-type-1-receptor (AT-II-1) antagonist, already a widely established antihypertensive agent in daily clinical practice, was proven to provide multiple neuroprotective effects in patients suffering ischemic and haemorrhagic stroke such as SAH [2]. Furthermore, an impaired ET-B_1_-R vasorelaxation was enhanced and restored under normal physiological conditions as well as after SAH.

Therefore, we aimed to analyse the effects of ET-1 in the presence of prostaglandins under normal physiological conditions, as well as of the ET-B_1_-R agonist Sarafotoxin S6c in the presence of prostaglandins under normal physiological and pathophysiological conditions, like SAH, to confirm the already well-known impaired vasorelaxation and ET-B_1_-R-mediated effect after SAH [6]. Furthermore, we aimed to characterise a possible dose-dependent effect of LOS on ET-1- and 5-HT-mediated vasoconstriction.

## 2. Materials and Methods

### 2.1. Ethical Approval and Animal Housing

All experimental procedures were approved by the local ethics committee of Darmstadt, Germany (Gen. Nr. F 138/12, 28 December 2009). The ARRIVE guidelines were strictly followed and experimental procedures complied with Directive 2010/63/UE [10]. Only male Sprague Dawley rats weighting between 310−422 g (mean 331.41 g ± 15.21 g) were included for further experimental procedures after fulfilling an acclimatisation period of 1 week. A maximum of five rats were kept in a cage prior to surgery, combined with a 12-h dark-light cycle. Food and water intake were possible ad libitum.

### 2.2. Randomisation

All rats were randomised into either a sham or SAH group. Blinding regarding these two groups was feasible neither for the surgeon nor the animal care takers, due to neurological deficits being obvious in the SAH group.

### 2.3. Anaesthesia Protocol and Postoperative Care

Anaesthesia was induced via an intraperitoneal injection of midazolam (1 mg/kg body weight) and ketamine (100 mg/kg body weight). During the whole operative procedure, animals breathed spontaneously without supplementary oxygen provided. Postoperatively, animals recovered in single cages. Rescue analgesia was provided by administration of 5 mL crystalloid solution with 0.0125 mg fentanyl intraperitoneal at least twice daily. Neurological deficits were objectified using the Bederson scale each day until euthanasia.

### 2.4. Surgical and Terminal Protocol

Regarding all detailed surgical steps, we refer to previously published data [11]. In brief, after anaesthesia, all rats were placed in a stereotactic frame with the head fixated via two pins in the internal acoustic meatus. The atlantoocipital membrane was opened and a first SAH was induced via application of autologous blood. The same surgical procedure was performed 24 h later. Previous to the first blood induction, the femoral vein was carefully dissected free, and a small catheter induced to gain access to the blood stream. The catheter stayed in place until day 5 post–blood induction, prior to euthanasia. In the sham group, 0.9% isotonic sodium chloride solution was injected instead of autologous blood. Before injection of 0.1 mL blood or sodium chloride solution, 0.2 mL cerebrospinal fluid was withdrawn. Severity of bleeding, neurological deficits, angiographic detected vasospasm, and magnetic resonance perfusion deficits have already been published elsewhere [11,12].

On day 5, after the second sodium chloride injection, and on day 5 postoperatively, after experimentally induced SAH, all rats were anesthetised via CO_2_ narcosis. Directly afterwards, each rat was exsanguinated by cutting both the external and internal carotid arteries. The brain, along with the cerebral vessels, were meticulously excised and immediately immersed in a cold modified Krebs-Högestätt solution. This solution was freshly prepared on the day of each experiment. Basal surface of the brain was macroscopically inspected for signs of SAH in each SAH animal.

The basilar artery, along with a small subarachnoid plane, were carefully dissected from the brainstem with the use of a binocular microscope (Carl Zeiss, Germany). Each basilar artery was cut into four equal parts measuring 2 mm. Basilar artery segments of every rat were randomly assigned to the different experimental groups, enabling the use of vessel segments from different animals in the final analysis for each group. Therefore, multiple vessel segments from each rat were taken. For analysis, 86 sham and 25 SAH vessel segments were included.

### 2.5. Experimental Setting

Every ring segment was mounted on stainless steel rods and introduced to an organ bath (IOA-5301; FMI GmbH, Germany). Changes of isometric force were registered in millinewton and quantified using a transducer system (GM Scaime, Annemasse Cedex, France).

Each organ bath was filled with a modified Krebs-Högestätt solution. Continuous carbonatisation was guaranteed through applying a humidified gas mixture (95% O_2_, 5% CO_2_). Measures resulted in a pH-value of approximately 7.35. 

Vessel segments without LOS were defined as control group. A reference contraction was induced by 124 mM potassium + Krebs solution (124 mM KCl) (Krebs-Högestätt solution with equimolar exchange of sodium chloride (NaCl) by KCl) and repeated at the end of each experiment. Vessel segments reaching less than 2 millinewton of contraction were excluded before ongoing experimental procedures. Segments developing less than 75% of the initial reference contraction at the end of the experiment were also excluded from further analysis. The functional integrity of the endothelium was tested by application of acetylcholine (10^−4^ M) after precontraction with 5-HT (10^−5^ M). A vasorelaxation of more than 30% indicated a functionally intact endothelium. Segments not reaching this reference percentage were excluded from further experiments.

AL-8810, a prostaglandin analogue, was applied with a concentration of 10^−5^ M. After an incubation period of 30 min, a cumulative ET-1 series (10^−11^ M, 10^−10^ M, 3 × 10^−10^ M, 10^−9^ M, 3 × 10^−9^ M, 10^−8^ M, 3 × 10^−8^ M, 10^−7^ M, 3 × 10^−7^ M) was applied in sham-group operated animals. A waiting time of 5 min was implemented prior to pipetting the next higher concentration. ET-1 concentrations were chosen by adhering to prior published data of our group [5,13,14].

AL-8810 was also applied with a concentration of 10^−5^ M in sham and SAH segments after prior precontraction with 62 mM KCl. Incubation time was 30 min for AL-8810. Afterwards, vessel segment relaxation was analysed by cumulative application of a Sarafotoxin S6c series (10^−14^ M, 10^−13^ M, 10^−12^ M, 3 × 10^−12^ M, 10^−11^ M, 3 × 10^−11^ M, 10^−10^ M, 3 × 10^−10^ M, 10^−9^ M). The waiting period between applications of the next higher concentration was 15 min.

LOS was administered in concentrations of 10^−3^ M, 3 × 10^−4^ M, 10^−4^ M, 10^−5^ M, 10^−6^ M, or as solvent control. Each vessel segment was preincubated for 30 min before a cumulative ET-1 series (10^−11^ M, 10^−10^ M, 3 × 10^−10^ M, 10^−9^ M, 3 × 10^−9^ M, 10^−8^ M, 3 × 10^−8^ M, 10^−7^ M, 3 × 10^−7^ M) was applied. Waiting time between administering the next higher ET-1 concentration was 5 min. In each case, only one LOS concentration effect-curve (CEC) was conducted by cumulative application to avoid tachyphylaxis.

LOS was moreover administered in a concentration of 10^−3^ M, 3 × 10^−4^ M, 10^−4^ M and 10^−5^ M, with a preincubation period of 30 min. Afterwards, a cumulative 5-HT series (10^−9^ M, 10^−8^ M, 10^−7^ M, 10^−6^ M, 10^−5^ M) was applied. A waiting period of 5 min was implemented prior to the next higher 5-HT application. In each case, only one LOS CEC was conducted by cumulative application to avoid tachyphylaxis. For the vasoconstrictive 5-HT and ET-1 series, a minimum 5-min waiting period was implemented. If vasoconstriction was still ongoing, the next plateau was awaited, and in cases of persisting increase in isometric force, the next concentration was given after 15 min.

### 2.6. Compounds and Solvents

Krebs-Högestätt solution was composed of the following components: NaCl (Sigma Aldrich, Darmstadt, Germany), 119 mM; KCl (Sigma Aldrich, Germany), 3.0 mM; sodium dihydrogen phosphate, 1.2 mM (AppliChem, Darmstadt, Germany); calcium chloride, 1.5 mM (AppliChem, Germany); magnesium chloride (Merck, Darmstadt, Germany), 1.2 mM; sodium hydrogen phosphate (VWR International BVBA, Leuven, Belgium), 15 mM; and glucose (Sigma Aldrich, Germany), 10 mM.

Acetylcholine, 5-HT, AL-8810, LOS, S6c, and ET-1 were acquired from Sigma-Aldrich (Schnelldorf, Darmstadt, Germany). All compounds were directly dissolved in distilled water on the day of each experiment. The selectivity of these compounds was described earlier [5,6].

### 2.7. Analysis of Results and Statistics

Vasocontraction was measured in millinewton and given as a percentage of the reference contraction. All values in the text and figures are given as mean ± standard deviation. For each completed CEC, the E_max_ (maximal vasorelaxation or contraction) and EC_50_ (the concentration at which half of the maximal effect occurs) were calculated. All statistical analyses, except for the vasorelaxative sham and SAH investigations with prior precontraction by 62 mM KCl and AL-8810 10^−5^ M, where the non-parametric Wilcoxon-Mann-Whitney test was used, were performed using the one-way analysis of variance followed by Scheffe’s test for post hoc comparisons of mean values. Probability values (*p*) less than 0.05 and 0.01 were considered significant. Data were analysed using IBM SPPS^®^ (version 22, Armonk, NY, USA). The sample size per group was determined using an a priori sample size calculation (BiAS.for.Windows–^®^ Version 11, epsilon Verlag, Nordhastedt, Germany). To achieve α = 0.05 at β = 0.2 with a sigma of 0.2, the sample size calculation showed that *n* = 4–8 segments per group was appropriate to have a delta between 0.3 to 0.5. Figures were created using Microcal Origin 7.0 (OriginLab, Northampton, MA, USA).

## 3. Results

In total, 111 basilar artery ring segments were included (86 sham segments, 25 SAH segments), corresponding to 28 rats in all. Finally, 76 sham and 10 SAH segments were analysed (achieving a sample size of at least five vessel segments per subgroup).

### 3.1. Effect of AL-8810 10^−5^ M on ET-1-Induced Vasocontraction in Sham Segments

In sham segments, the E_max_ elicited by ET-1 was 76%. A dose-dependent vasoconstriction from all vessel segments preincubated with ET-1 (10^−11^ M to 3 × 10^−7^ M) was observed (Figure 1, Table 1). The calculated pD_2_ was 7.44.

### 3.2. Effect of 62 mM KCl and AL-8810 10^−5^ M Precontraction on Sarafotoxin S6c-Induced Vasorelaxation in Sham and SAH Segments

All vessel segments were precontracted by application of 62 mM KCl and AL-8810 10^−5^ M. In sham and SAH segments, a Sarafotoxin S6c series (10^−14^–10^−9^ M) was performed. In all sham segments, Sarafotoxin S6c induced a vasorelaxation (E_max_: 47%) (Figure 2, Table 2).

SAH day 5 segments (E_max_: 38%) showed a weaker vasorelaxation compared to the sham group. The maximal vasorelaxation between the sham and SAH day 5 group was non-significantly altered (*p* = 0.18). In sham segments, administration of Sarafotoxin S6c 10^−9^ M showed a statistically enhanced vasorelaxation (44%, *p* = 0.04) as compared to the SAH day 5 group (28%). 10^−14^ M (*p* = 0.06) and 3 × 10^−12^ M (*p* = 0.06) Sarafotoxin S6c in the sham group showed a non-significant trend towards an enhanced vasorelaxation compared to the SAH day 5 group (Figure 2, Table 2).

### 3.3. Effect of LOS on ET-1-Induced Vasocontraction in Sham Segments 

All vessel segments were preincubated with LOS in different concentrations (10^−3^ M, 3 × 10^−4^ M, 10^−4^ M, 10^−5^ M, 10^−6^ M, solvent control) followed by different ET-1 concentrations applied in a cumulative fashion (ET-1 10^−11^ M to 3 × 10^−7^ M). LOS 10^−3^ M elicited an E_max_ of 15%, LOS 3 × 10^−4^ M with 44%, LOS 10^−4^ M with 73%, LOS 10^−5^ M with 89%, LOS 10^−6^ M with 105%, and the solvent control group elicited an E_max_ of 122% (Figure 3, Table 3).

E_max_: vasocontraction was significantly reduced in the LOS 10^−3^ M group compared to the 10^−5^ M (*p* = 0.004), 10^−6^ M (*p* = 0.0001) and solvent control group (*p* = 0.000005). Furthermore, vasoconstriction was significantly reduced in the LOS 3 × 10^−4^ M group compared to 10^−6^ M (*p* = 0.003) and solvent control (*p* = 0.0001). Furthermore, vasocontraction was significantly reduced in the LOS 10^−4^ M group compared to the solvent control group (*p* = 0.03).

Significant changes in the pD_2_ were not noted throughout all concentrations, suggesting the induced contraction seemed to be altered in a non-competitive fashion.

### 3.4. Effect of LOS on 5-HT-Induced Vasocontraction in Sham Segments

All vessel segments were preincubated with LOS in different concentrations (10^−3^ M, 3 × 10^−4^ M, 10^−4^ M, 10^−5^ M) followed by application of different 5-HT concentrations in a cumulative fashion (10^−9^ M to 10^−5^ M). An E_max_ for LOS 10^−3^ M was detected with 10%, for 3 × 10^−4^ M with 75%, for 10^−4^ M with 82%, and 10^−5^ M with 84% (Figure 4, Table 4).

E_max_: maximal vasoconstriction was significantly reduced in the LOS 10^−3^ M group compared to the LOS 3 × 10^−4^ M group (*p* = 0.00002), 10^−4^ M group (*p* = 0.000006), and 10^−5^ M group (*p* = 0.000004).

The pD_2_ values were significantly altered comparing the LOS 10^−3^ M group to 3 × 10^−4^ M (*p* = 0.03), LOS 3 × 10^−4^ M (*p* = 0.04), and 10^−5^ M group (*p* = 0.02), indicating that the induced contraction seemed to be altered in a competitive fashion.

## 4. Discussion

To the best of the authors’ knowledge, our experimental findings demonstrate for the first time that LOS applied in a concentration of 10^−3^ M possesses the ability to diminish experimentally induced DCVS in a proinflammatory microenvironment, created by the application of 5-HT and ET-1, both key players in the development of cerebral vasospasm, in sham segments. In addition to a concentration-dependent antagonisation of an already known ET-1-induced vasocontraction, a 5-HT-mediated vasoconstriction of basilar artery ring segments was antagonised in a dose-dependent manner [9].

In cerebrovasculature, an ET-A-R can be found on the muscular media besides an ET-B_1_-R, located on the endothelium. Under normal physiological conditions, the ET-A-R mediates vasoconstrictive effects, which normally masks an ET-B_1_-R-mediated vasorelaxation. After SAH, besides an elevated ET-1 synthesis, as well as an upregulated ET-A-R expression, the vasorelaxative ET-B_1_-R is functionally impaired [2,9]. These aspects promote macro as well as micro DCVS.

Likewise, under normal physiological circumstances, an AT-II-1 receptor is also located on the muscular vessel media, mediating vasoconstriction, whereas an AT-II-2 receptor is located on the endothelium triggering vasorelaxation [15]. After SAH, vasocontractile AT-II-1 receptors and 5-HT receptors on the muscular media are overexpressed, as are vasocontractile peptides like 5-HT [15,16]. Findings of enhanced 5-HT release have already been described by Lobato and Roman et al., analysing its vasocontractile role in an SAH cat model and in diabetic rats [16,17]. Furthermore, Ansar et al. highlighted the correlation of 5-HT receptor upregulation with reducing regional cerebral blood flow after SAH [15]. Sercombe et al. furthermore demonstrated the important role of 5-HT, increasingly released after SAH, in mediating neurogenic inflammation by disrupting the blood-brain barrier. Besides 5-HT, substance P is also involved in causing blood-brain barrier disruption and degranulation of mast cells with consecutive leukocyte activation, therefore mediating neuroinflammation in an indirect fashion [18]. Our study findings—in which a microinflammatory environment was created by the application of 5-HT and AL-8810—indicate a dose-dependent and strong antagonising effect of LOS on 5-HT mediated vasoconstriction in sham segments. For LOS 10^−3^ M, a significantly reduced E_max_ of 10% was observed, whereas for LOS 3 × 10^−4^ M, the maximal vasoconstriction was still as high as 75% (*p* = 0.00002). For LOS 10^−4^ M (*p* = 0.000006) and 10^−5^ M (*p* = 0.000004), E_max_ was detected at 82% and 84%, respectively, and was significantly enhanced compared to the LOS 10^−3^ M group.

Furthermore, it is already well known that, under normal physiological and pathophysiological circumstances after experimental induced SAH, LOS antagonises an ET-1-mediated cerebral vasoconstriction [9]. After SAH, the ET-B_1_-R function is normally impaired [6], and after LOS administration, beneficial effects in restoring this vasorelaxative receptor function have been observed. As demonstrated, regarding the effects of LOS on ET-A-R-induced contraction and ET-B_1_-R-induced relaxation, in our series ET-1 induced a dose-dependent increase in vasocontraction, which has been significantly reduced after preincubation with LOS 10^−3^ M without changes of the pD_2_. Therefore, LOS elicited a diminished vasocontraction, but without affecting the shift of the CEC, so that this antagonism seems to be a non-competitive one. A similar effect has already been analyzed in prior series [9]. Our group points out that this reduced vasocontraction was abolished after preincubation with BQ-788, an ET-B_1_-R antagonist, but not after preincubation with BQ-123, an ET-A-R antagonist. Therefore, it seems obvious that this mechanism seems to be ET-B_1_-R-dependent. Facing the impaired vasorelaxatory ET-B_1_-R function after SAH, a non-competitive beneficial influence restoring ET-B_1_-R mediated vasorelaxativity could be a good therapeutic approach to reduce DCVS after SAH [9].

In terms of LOS-mediated effects on the ET-B_1_-R-dependent pathway, Sarafotoxin S6c applied in our series showed increased relaxation and higher receptor sensitivity in the presence of LOS, significantly for 10^−9^ M. An increased relaxation and significantly higher sensitivity through Sarafotoxin SC6 under preincubation with LOS has also observed in other series [9]. Therefore, these data suggest that LOS as AT-II-1 antagonist possesses a positive modulatory effect on the ET-B_1_-R. Activation of the ET-B_1_-R causes an increased relaxation via the NO-cGMP pathway. A detailed description regarding the influence of LOS on this pathway has already been analyze elsewhere [9].

Furthermore, it has been clearly demonstrated that under selective ET-A- and ET-B_1_-R blockage, LOS delivers its effect via a non-competitive ET-B_1_-R antagonism [9]. Our data clearly show that LOS 10^−3^ M provides a strong and dose-dependent antagonism on ET-1-induced vasocontraction resulting in an E_max_ of 15% compared to LOS 10^−5^ M (E_max_: 89%; *p* = 0.004), 10^−6^ M (E_max_: 105%; *p* = 0.0001), and solvent control group (E_max_: 122%; *p* = 0.000005). These findings are aligned with the current literature. 

Besides potential sequelae, in terms of DCVS with delayed cerebral ischemia and ischemic neurological deficits, cortical spreading depressions and cerebral inflammation are well-known epiphenomena after SAH [19,20,21]. In addition to the confirmed spasmolytic effects of LOS, interestingly, its administration in a rat epilepsy model was shown to diminish epileptogenic activity and neuronal breakdown by exerting neuroprotective effects in the CA1 hippocampal regions [19,21,22]. Regarding cerebral inflammation, significant evidence exists that after SAH, besides enhanced 5-HT synthesis, prostaglandins as prostaglandin F2aplha are also upregulated [3,8]. Upregulated concentrations of eicosanoids can be found in cerebrospinal fluid and are already known to participate in the pathogenesis of DCVS [3,8]. Furthermore, the literature reveals that LOS was already proven to antagonise a prostaglandin F2a-mediated vasoconstriction, leading to significant vasorelaxation in precontracted vessels under normal physiological conditions and after SAH [9]. Our data showed that even under precontraction with the prostaglandin analogue AL-8810, ET-1 application in a cumulative fashion still elicited a strong vasoconstriction. Furthermore, our data demonstrated that 62 mM KCl and AL-8810 precontracted vessels showed a vasorelaxation on Sarafotoxin S6c, an ET-B_1_-R agonist, that mediates vasorelaxation via the acetylcholine, cGMP, and NO-pathway. Recent findings clearly demonstrated a significant activation of the vasorelaxant subparts of the ET-B_1_-R pathway for Sarafotoxin S6c, cGMP, and sodium nitroprusside after SAH compared to a corresponding solvent control group. In our series, an enhanced E_max_ in vasorelaxation was observed comparing the solvent control group to SAH day 5 group (E_max_ 47% vs. 38%, respectively; *p* = 0.18). These findings are intriguing, regarding the already known impaired ET-B_1_-R function after SAH [6]. Sarafotoxin SC6 10^−9^ M induced a significantly enhanced vasorelaxation in the sham group compared to the SAH day 5 group (*p* = 0.04). Potentially, if administering LOS in this setting in the SAH group, vasorelaxative effects will also be exerted by reactivating the vasorelaxative ET-B_1_-R subpart. To restore the ET-B_1_-R function through LOS in a non-competitive fashion, consideration might be given to the effect of Sarafotoxin SC6 under a microinflammatory environment created in the presence of AL-8810.

In addition, regarding an enhanced AT-II-1 receptor expression post-SAH [15,23], a LOS application might be beneficial by directly antagonising the AT-II-1 receptor in a competitive fashion as well. Next to DCVS, delayed cerebral ischemia, delayed ischemic neurological deficits, and cerebral inflammation, poor patient outcomes after SAH is a multifactorial process depicted by neurodegeneration, early elevation of intracranial pressure, cortical spreading depressions and microcirculatory disturbances, and impairment of cerebral autoregulation.

### 4.1. Clinical Context

LOS is an already well-established clinical drug commonly used for antihypertensive treatment and heart failure therapy. Therefore, it is widely available throughout the continent. Regarding all the positive effects, which have already been observed after ischemic and haemorrhagic stroke in preclinical and clinical trials [2], the question arises as to whether there might be room for neuroprotection even after SAH. This is especially important since its application does not influence global cerebral perfusion pressure in essential hypertonic patients, which can be set equivalent to the needed-hypertension in patients suffering SAH. This question might be difficult to answer in a retrospective fashion because the standard of care remains that all antihypertensive agents, after the initial bleeding event, are discontinued to avoid hypotensive episodes with potential irreversible delayed ischemic neurological deficits during the critical phase of DCVS. Additionally, it is vague to postulate that the effect of LOS is long-lasting over the phase of DCVS after discontinuing it upon patient admission.

### 4.2. Study Limitations

The main limitation of this work is that the authors analysed only in vitro tension studies, resulting in the inability to dig deeper in the multifactorial process potentially involved in poor patient outcomes after SAH (microspasm, impairment of the cerebral autoregulation, spreading depressions, or analysing other compartments as different concentrations of 5-HT, ET-1, or prostaglandins in the blood or cerebrospinal fluid).

## 5. Conclusions

Overall, LOS is a potentially underrated neuroprotective drug with certain advantages when given to patients suffering aneurysmal SAH. Its effects have been studied in a rat model only, without consideration for the multifactorial process responsible for poor patient outcomes after SAH. LOS ameliorates DCVS through the restoration of the ET-B1-R function, provides antiepileptic effects, antagonises 5-HT-mediated cerebral inflammation, and improves neuroregeneration. Therefore, physicians might think outside the box and prospective randomised controlled trials are highly warranted.

## Figures and Tables

**Figure 1 jcm-11-07367-f001:**
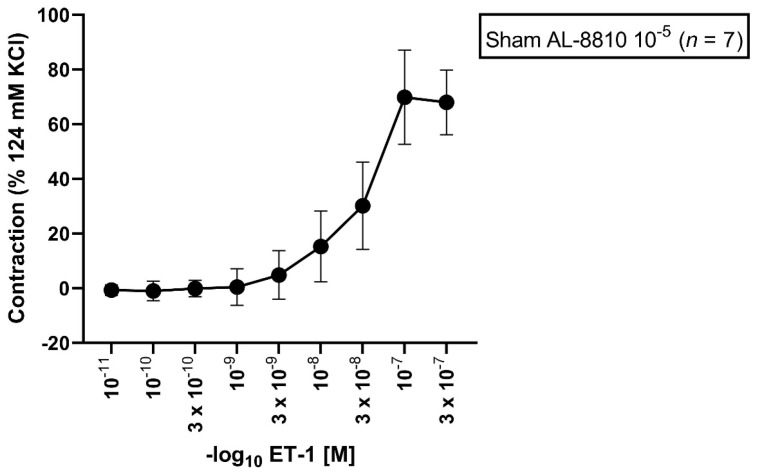
Effect of ET-1-induced vasocontraction by prior preincubation with AL-8810 10^−5^ M in sham vessel segments. ET-1 elicited a dose-dependent vasoconstriction. Shown is the CEC for the group with AL-8810 10^−5^ M. ET-1 triggered a dose-dependent vasoconstriction in sham segments.

**Figure 2 jcm-11-07367-f002:**
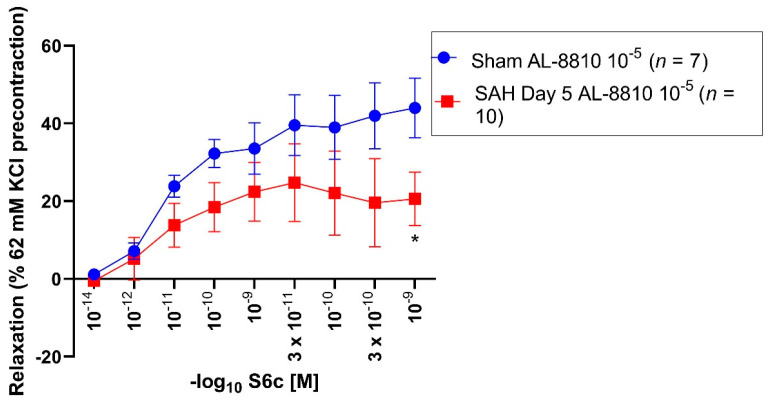
Vasorelaxative effect of Sarafotoxin S6c after precontraction with 62 mM KCl and AL-8810 10^−5^ M in sham and SAH day 5 segments. Sarafotoxin S6c elicited a dose-dependent vasorelaxation. Shown are the CECs for each group with AL-8810 10^−5^ M. Sarafotoxin S6c triggered a dose-dependent relaxation in the control and SAH day 5 groups. The E_max_ was predominantly reduced in the control group (E_max_ 47%) compared to the SAH day 5 group (E_max_ 38%). For Sarafotoxin S6c 10^−9^ M, the vasoconstriction was significantly reduced in the control group compared to the SAH day 5 group (*p* = 0.04). * *p* < 0.05.

**Figure 3 jcm-11-07367-f003:**
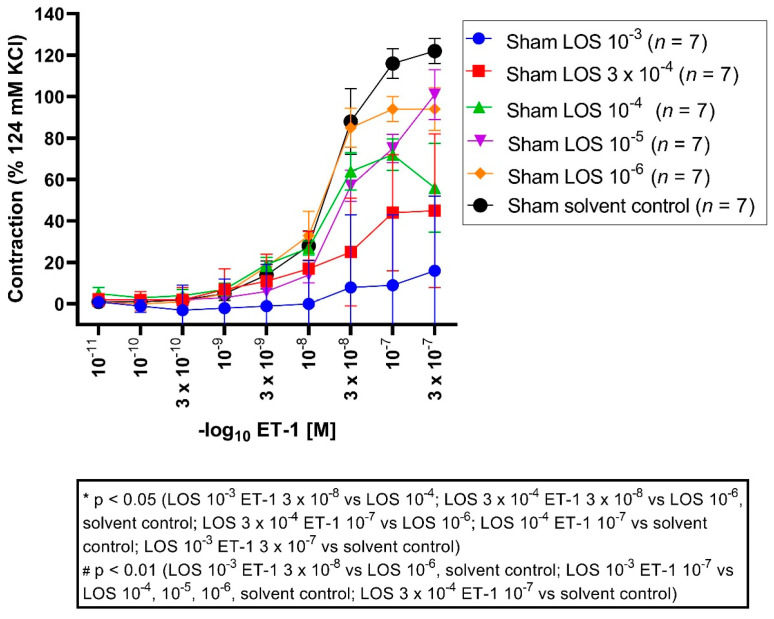
Effect of LOS on ET-1-induced vasocontraction in sham segments. ET-1 elicited a dose-dependent vasoconstriction. Shown are the CECS for each group with LOS (10^−3^ M, 3 × 10^−4^ M, 10^−4^ M, 10^−5^ M, 10^−6^ M, solvent control). LOS reduced the E_max_ dose dependent, so that the vasoconstrictive effects for ET-1 were stronger in the lower molar groups compared to the higher ones (E_max_ LOS 10^−3^ M: 15%, E_max_ LOS 3 × 10^−4^ M: 44%, E_max_ LOS 10^−4^ M: 73%, E_max_ LOS 10^−5^ M: 89%, E_max_ LOS 10^−6^ M: 105%, E_max_ solvent control: 122%). For ET-1 3 × 10^−8^ M, vasocontraction was significantly reduced in the LOS 10^−3^ M group compared to the 10^−4^ M group (*p* = 0.04), compared to the 10^−6^ M group (*p* = 0.001) and solvent control group (*p* = 0.001), as well for the LOS 3 × 10^−4^ M group compared to the 10^−6^ M group (*p* = 0.02) and solvent control group (*p* = 0.01). For ET-1 10^−7^ M, vasoconstriction was also reduced comparing LOS 10^−3^ M with 10^−4^ M (*p* = 0.001), 10^−5^ M (*p* = 0.0007), 10^−6^ M (*p* = 0.00001), and with a solvent control group (*p* = 0.0000), as well for LOS 3 × 10^−4^ M compared to 10^−6^ M (*p* = 0.01) and to a solvent control group (*p* = 0.0001). Vasocontraction was also significantly reduced comparing LOS 10^−4^ M to a solvent control group (*p* = 0.04). For ET-1 3 × 10^−7^ M, LOS 10^−3^ M showed a significantly reduced contraction compared to the solvent control group (*p* = 0.01). * *p* < 0.05, # *p* < 0.01.

**Figure 4 jcm-11-07367-f004:**
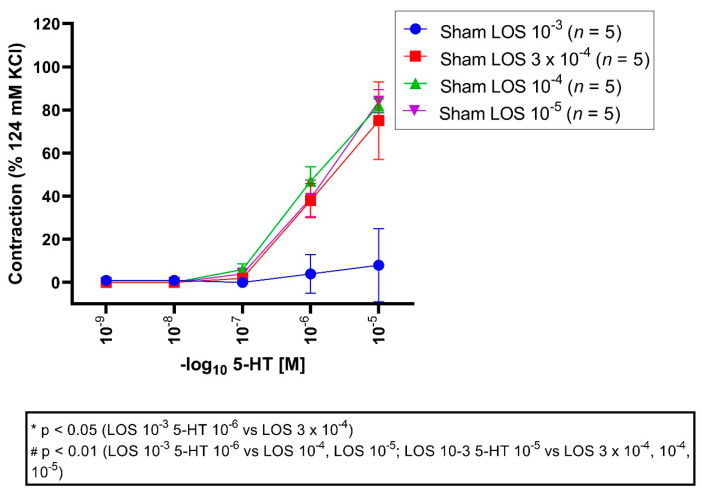
Effect of LOS on 5-HT-induced vasocontraction in sham segments. 5-HT elicited a dose-dependent vasocontraction. Shown are the CECs for each group with LOS. LOS reduced the E_max_ dose dependent. 5-HT triggered a dose-dependent contraction in the sham groups. An E_max_ of 10%, 75%, 82%, and 84% was depicted for LOS 10^−3^ M, 3 × 10^−4^ M, 10^−4^ M and 10^−5^ M groups, respectively. For 5-HT 10^−6^ M and 10^−5^ M, vasocontractions were significantly reduced in the LOS 10^−3^ group compared to 3 × 10^−4^ M (5-HT 10^−6^ M, *p* = 0.01; 5-HT 10^−5^ M, *p* = 0.00001), 10^−4^ M (5-HT 10^−6^ M, *p* = 0.001; 5-HT 10^−5^ M, *p* = 0.000006), and 10^−5^ M group (5-HT 10^−6^ M, *p* = 0.008; 5-HT 10^−5^ M, *p* = 0.000004). * *p* < 0.05, # *p* < 0.01.

**Table 1 jcm-11-07367-t001:** Contractile assessment after incubation with AL-8810 10^−5^ M followed by an ET-1 series under normal physiological conditions. ET-1 (10^−11^–3 × 10^−7^ M) induced a dose-dependent vasoconstriction. Values are expressed as mean ± standard deviation.

	ET-1 Series	E(max) Contraction	pD_2_	*n*
Sham + AL-8810 10^−5^ M		76% ± 13%	7.44 ± 0.18	7

**Table 2 jcm-11-07367-t002:** Vasorelaxative assessment after preincubation with 62 mM KCl and AL-8810 10^−5^ M in sham and SAH day 5 segments. Sarafotoxin S6c induced a more pronounced relaxation in sham segments compared to SAH day 5 segments. Values are expressed as mean ± standard deviation.

	Sarafotoxin S6c Series	E(max) Relaxation	pD_2_	*n*
Sham + 62 mM KCl + AL-8810 10^−5^ M		47% ± 15%	12.06 ± 0.42	7
SAH day 5 + 62 mM KCl + AL-8810 10^−5^ M		38% ± 22%	12.02 ± 0.63	10

**Table 3 jcm-11-07367-t003:** Vasocontraction assessment after ET-1 administration and prior preincubation with LOS in different concentrations in sham segments. ET-1 induced a dose-dependent vasoconstriction, which was dose-dependent antagonised by LOS (10^−3^ M, 3 × 10^4^ M, 10^−5^ M, 10^−6^ M). Values are expressed as mean ± standard deviation and compared to the solvent control group. * *p* < 0.05 vs. LOS 10^−4^ M, # *p* < 0.01 vs. LOS 3 × 10^−4^ M, # *p* < 0.01 vs. LOS 10^−3^ M.

	ET-1 Series	E(max) Contraction	pD2	*n*
Sham + LOS 10^−3^ M		15% ± 31% #	7.91 ± 2.00	7
Sham + LOS 3 × 10^−4^ M		44% ± 28% #	7.54 ± 0.54	7
Sham + LOS 10^−4^ M		73% ± 22% *	7.52 ± 0.37	7
Sham + LOS 10^−5^ M		89% ± 14%	7.10 ± 0.31	7
Sham + LOS 10^−6^ M		105% ± 8%	7.33 ± 0.35	7
Sham + solvent control		122% ± 15%	7.30 ± 0.13	7

**Table 4 jcm-11-07367-t004:** Vasocontraction assessment after 5-HT administration and prior preincubation with LOS in different concentrations (LOS 10^−3^ M, LOS 3 × 10^−4^ M, LOS 10^−4^ M, LOS 10^−5^ M) in sham segments. 5-HT induced a dose-dependent vasocontraction, which was also dose-dependent reversed by LOS. Values are expressed as mean ± standard deviation and compared to the LOS 3 × 10^−4^ M, 10^−4^ M, 10^−5^ M groups. * *p* < 0.05; # *p* < 0.01 vs. LOS 3 × 10^−4^ M, # *p* < 0.01 vs. LOS 10^−4^ M, # *p* < 0.01 vs. LOS 10^−5^ M.

	5-HT Series	E(max) Contraction	pD2	*n*
Sham + LOS 10^−3^ M		10% ± 16%	7.54 ± 1.42	5
Sham + LOS 3 × 10^−4^ M		75% ± 18% #	6.04 ± 0.08 *	5
Sham + LOS 10^−4^ M		82% ± 7% #	6.11 ± 0.27 *	5
Sham + LOS 10^−5^ M		84% ± 12% #	5.95 ± 0.26 *	5

## Data Availability

All data supporting the reported results are published within this manuscript.

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
