# Peer review of "The Effect of Losartan on Neuroinflammation as Well as on Endothelin-1- and Serotonin-Induced Vasoconstriction in a Double-Haemorrhage Rat Model"

_jcm, 2022, doi:10.3390/jcm11247367_

Round 1
Reviewer 1 Report
the authors demonstrate the role of Losartan in aneurysmal subarachnoid hemorrhage using an animal model. The study design has been described in very detail, while the mechanism and the pathway are still vague. We are looking forward to furthering discussion on this part.
Author Response
Reviewer 1
the authors demonstrate the role of Losartan in aneurysmal subarachnoid hemorrhage using an animal model. The study design has been described in very detail, while the mechanism and the pathway are still vague. We are looking forward to furthering discussion on this part.
REPLY: We cordially thank reviewer 1 for commenting on our experimental study. We aimed to adapt our discussion as follows from:
“Our findings demonstrate that LOS possesses the ability to diminish experimentally induced DCVS in a proinflammatory microenvironment created by the application of 5-HT and ET-1 in sham segments. Besides a concentration-dependent antagonisation of an already known ET-1-induced vasocontraction, a 5-HT-mediated vasoconstriction of BA ring segments was antagonised as well in a dose-dependent manner 9, even stronger with the application of LOS 10-3 M.
In cerebrovasculature, an ET-A-R can be found on the muscular media besides an ET-B1-R, located on the endothelium. Under normal physiological conditions, the ET-A-R mediates vasoconstrictive effects, which normally mask an ET-B1-R-mediated vasorelaxation . After SAH, besides an elevated ET-1 synthesis, as well as an upregulated ET-A-R expression, the vasorelaxative ET-B1-R is functionally impaired 2,9. These aspects promote macro as well as micro DCVS.
Likewise, under normal physiological circumstances, an AT-II-1 receptor is also located on the muscular vessel media, mediating vasoconstriction, whereas an AT-II-2 receptor is located on the endothelium triggering vasorelaxation 15. After SAH, vasocontractile AT-II-1 receptors as well as 5-HT receptors on the muscular media are overexpressed, as are vasocontractile peptides like 5-HT 15,16. Findings of enhanced 5-HT release have already been described by Lobato and Roman et al., analysing its vasocontractile role in an SAH cat model and in diabetic rats 16,17. Even more, Ansar et al. highlighted the correlation of 5-HT receptor upregulation with reducing regional cerebral blood flow after SAH 15. Sercombe et al. furthermore demonstrated the important role of 5-HT, increasingly released after SAH, in mediating neurogenic inflammation by disrupting the blood brain barrier. Besides 5-HT, substance P is also involved in causing blood brain barrier disruption and degranulation of mast cells with consecutive leukocyte activation, therefore mediating neuroinflammation in an indirect fashion 18. Our study findings—in which a microinflammatory environment was created by the application of 5-HT and AL-8810—indicate a dose-dependent and strong antagonising effect on 5-HT mediated vasoconstriction in sham segments. For LOS 10-3 M, a significantly reduced Emax of 10% was observed, whereas for LOS 3 x 10-4 M, the maximal vasoconstriction was still as high as 75% (p = 0.00002). For LOS 10-4 M (p = 0.000006) and 10-5 M (p = 0.000004), Emax was detected by 82% and 84%, respectively, and was significantly enhanced compared to the LOS 10-3 M group.
Furthermore, it is already well-known, that under normal physiological as well as pathophysiological circumstances after experimental induced SAH, LOS antagonises an ET-1-mediated cerebral vasoconstriction 9. After SAH, the ET-B1-R function is normally impaired 6, and after LOS administration, beneficial effects in restoring this vasorelaxative receptor function have been observed . Furthermore, it has been clearly demonstrated that under selective ET-A- and ET-B1-R blockage, LOS delivers its effect via a non-competitive ET-B1-R antagonism 9. Our data clearly show that LOS 10-3 M provides a strong and dose-dependent antagonism on ET-1-induced vasocontraction resulting in an Emax of 15% compared to LOS 10-5 M (Emax: 89%; p = 0.004), 10-6 M (Emax: 105%; p = 0.0001), and solvent control group (Emax: 122%; p = 0.000005). These findings are aligned with current literature.
Besides potential sequelae, in terms of DCVS with delayed cerebral ischemia (DCI) and ischemic neurological deficits (DIND), CSD and CI are well-known epiphenomena after SAH 19-21. Besides confirmed spasmolytic effects of LOS, interestingly, its administration in a rat epilepsy model was shown to diminish epileptogenic activity and neuronal breakdown by exerting neuroprotective effects in the CA1 hippocampal regions 19,21,22. Regarding CI, a lot of evidence exists that after SAH, besides enhanced 5-HT synthesis, prostaglandins as PGF2a are also upregulated 3,8. Upregulated concentrations of eicosanoids could be found in CSF and are already known to participate in the pathogenesis of DCVS 3,8. Furthermore, literature reveals that LOS was already proven to antagonise a PGF2a-mediated vasoconstriction, leading to significant vasorelaxation in PGF2a precontracted vessels under normal physiological conditions and after SAH 9. Our data showed that even under precontraction with the prostaglandin analogue AL-8810, ET-1 application in a cumulative fashion still elicited a strong vasoconstriction. Furthermore, our data demonstrated that 62 mM KCl and AL-8810 precontracted vessels showed a vasorelaxation on S6c, an ET-B1-R agonist, that mediates vasorelaxation via the ACh, cGMP, and NO-pathway . Recent findings clearly demonstrated a significant activation of the vasorelaxant subparts of the ET-B1-R pathway for S6c, cGMP, and SNP after SAH compared to a corresponding solvent control group . In our series, an enhanced Emax in vasorelaxation was observed comparing the solvent control group to SAH day 5 (Emax 47% vs. 38%, respectively; p = 0.18). These findings seem intriguing, regarding the already known impaired ET-B1-R function after SAH 6. SC6 10-9 M induced a significantly enhanced vasorelaxation in the sham group compared to the SAH day 5 group (p = 0.04). So potentially, if administering LOS in this setting in the SAH group, vasorelaxative effects will also be exerted by reactivating the vasorelaxative ET-B1-R subpart. To restore the ET-B1-R function through LOS in a non-competitive fashion, consideration might be given to the effect of SC6 under a microinflammatory environment created in the presence of AL-8810.
Additively, regarding an enhanced AT-II-1 receptor expression post-SAH 15,23, a LOS application might be beneficial by directly antagonising the AT-II-1 receptor in competitive fashion, as well. Of note, next to DCVS, DCI, DIND, and CI, poor patient outcomes after SAH is a multifactorial process depicted by neurodegeneration, early elevation of ICP, CSD and microcirculatory disturbances, and impairment of cerebral autoregulation.
Clinical context
LOS is an already well-established clinical drug commonly used for antihypertensive treatment and heart failure therapy. Therefore, it is widely available throughout the continent. Regarding all the positive effects, which have already been observed after ischemic and haemorrhagic stroke in preclinical and clinical trials 2, the question arises if there might be a room for neuroprotection even after SAH. Even more, because its application does not influence global cerebral perfusion pressure in essential hypertonic patients, which can be set equipoise to needed hypertension in patients suffering SAH. This question might be difficult to answer in a retrospective fashion because the standard of care remains that all antihypertensive agents, after the initial bleeding event, are discontinued to avoid hypotensive episodes with potential irreversible DIND during the critical phase of DCVS. Also, it is vague to postulate that the effect of LOS is long-lasting over the phase of DCVS after discontinuing it upon patient admission.”
to
“Our findings demonstrate that LOS possesses the ability to diminish experimentally induced DCVS in a proinflammatory microenvironment created by the application of 5-HT and ET-1 in sham segments. Besides a concentration-dependent antagonisation of an already known ET-1-induced vasocontraction, a 5-HT-mediated vasoconstriction of BA ring segments was antagonised as well in a dose-dependent manner 9, even stronger with the application of LOS 10-3 M.
In cerebrovasculature, an ET-A-R can be found on the muscular media besides an ET-B1-R, located on the endothelium. Under normal physiological conditions, the ET-A-R mediates vasoconstrictive effects, which normally mask an ET-B1-R-mediated vasorelaxation. After SAH, besides an elevated ET-1 synthesis, as well as an upregulated ET-A-R expression, the vasorelaxative ET-B1-R is functionally impaired 2,9. These aspects promote macro as well as micro DCVS.
Likewise, under normal physiological circumstances, an AT-II-1 receptor is also located on the muscular vessel media, mediating vasoconstriction, whereas an AT-II-2 receptor is located on the endothelium triggering vasorelaxation 15. After SAH, vasocontractile AT-II-1 receptors as well as 5-HT receptors on the muscular media are overexpressed, as are vasocontractile peptides like 5-HT 15,16. Findings of enhanced 5-HT release have already been described by Lobato and Roman et al., analysing its vasocontractile role in an SAH cat model and in diabetic rats 16,17. Even more, Ansar et al. highlighted the correlation of 5-HT receptor upregulation with reducing regional cerebral blood flow after SAH 15. Sercombe et al. furthermore demonstrated the important role of 5-HT, increasingly released after SAH, in mediating neurogenic inflammation by disrupting the blood brain barrier. Besides 5-HT, substance P is also involved in causing blood brain barrier disruption and degranulation of mast cells with consecutive leukocyte activation, therefore mediating neuroinflammation in an indirect fashion 18. Our study findings—in which a microinflammatory environment was created by the application of 5-HT and AL-8810—indicate a dose-dependent and strong antagonising effect on 5-HT mediated vasoconstriction in sham segments. For LOS 10-3 M, a significantly reduced Emax of 10% was observed, whereas for LOS 3 x 10-4 M, the maximal vasoconstriction was still as high as 75% (p = 0.00002). For LOS 10-4 M (p = 0.000006) and 10-5 M (p = 0.000004), Emax was detected by 82% and 84%, respectively, and was significantly enhanced compared to the LOS 10-3 M group.
Furthermore, it is already well-known, that under normal physiological as well as pathophysiological circumstances after experimental induced SAH, LOS antagonises an ET-1-mediated cerebral vasoconstriction 9. After SAH, the ET-B1-R function is normally impaired 6, and after LOS administration, beneficial effects in restoring this vasorelaxative receptor function have been observed. As demonstrated regarding the effects of LOS on ET-A-R induced contraction and ET-B1-R induced relaxation, in our series ET-1 induced a dose-dependent increase in vasocontraction, which has been significantly reduced after preincubation with LOS 10-3 M without changes of the pD2. Therefore, LOS elicited a diminished vasocontraction, but without affecting the shift of the CEC, so that this antagonism seems to be a non-competitive one. A similar effect has already been analyzed in prior series.9 Here, our group pointed out, that this reduced vasocontraction was abolished after preincubation with BQ-788, an ET-B1-R antagonist, but not after preincubation with BQ-123, an ET-A-R antagonist. Therefore, it seems obvious that this mechanism seems to be ET-B1-R dependent. Facing the impaired vasorelaxatory ET-B1-R function after SAH, a non-competitive beneficial influence restoring ET-B1-R mediated vasorelaxativity could be a good therapeutic approach to reduce DCVS after SAH.9
In terms of LOS-mediated effects on the ET-B1-R dependent pathway, Sarafotoxin S6c applied in our series showed increased relaxation and higher receptor sensitivity in the presence of LOS, significantly for 10-9 M. An increased relaxation and significantly higher sensitivity through Sarafotoxin SC6 under preincubation with LOS has also observed in other series.9 Therefore, these data suggest that LOS as AT-II-1 antagonist possesses a positive modulatory effect on the ET-B1-R. Activation of the ET-B1-R causes an increased relaxation via the NO-cGMP pathway. A detailed description regarding the influence of LOS on this pathway has already been analyze elsewhere.9
Furthermore, it has been clearly demonstrated that under selective ET-A- and ET-B1-R blockage, LOS delivers its effect via a non-competitive ET-B1-R antagonism 9. Our data clearly show that LOS 10-3 M provides a strong and dose-dependent antagonism on ET-1-induced vasocontraction resulting in an Emax of 15% compared to LOS 10-5 M (Emax: 89%; p = 0.004), 10-6 M (Emax: 105%; p = 0.0001), and solvent control group (Emax: 122%; p = 0.000005). These findings are aligned with current literature.
Besides potential sequelae, in terms of DCVS with delayed cerebral ischemia (DCI) and ischemic neurological deficits (DIND), CSD and CI are well-known epiphenomena after SAH 19-21. Besides confirmed spasmolytic effects of LOS, interestingly, its administration in a rat epilepsy model was shown to diminish epileptogenic activity and neuronal breakdown by exerting neuroprotective effects in the CA1 hippocampal regions 19,21,22. Regarding CI, a lot of evidence exists that after SAH, besides enhanced 5-HT synthesis, prostaglandins as PGF2a are also upregulated 3,8. Upregulated concentrations of eicosanoids could be found in CSF and are already known to participate in the pathogenesis of DCVS 3,8. Furthermore, literature reveals that LOS was already proven to antagonise a PGF2a-mediated vasoconstriction, leading to significant vasorelaxation in PGF2a precontracted vessels under normal physiological conditions and after SAH 9. Our data showed that even under precontraction with the prostaglandin analogue AL-8810, ET-1 application in a cumulative fashion still elicited a strong vasoconstriction. Furthermore, our data demonstrated that 62 mM KCl and AL-8810 precontracted vessels showed a vasorelaxation on S6c, an ET-B1-R agonist, that mediates vasorelaxation via the ACh, cGMP, and NO-pathway . Recent findings clearly demonstrated a significant activation of the vasorelaxant subparts of the ET-B1-R pathway for S6c, cGMP, and SNP after SAH compared to a corresponding solvent control group . In our series, an enhanced Emax in vasorelaxation was observed comparing the solvent control group to SAH day 5 (Emax 47% vs. 38%, respectively; p = 0.18). These findings seem intriguing, regarding the already known impaired ET-B1-R function after SAH 6. SC6 10-9 M induced a significantly enhanced vasorelaxation in the sham group compared to the SAH day 5 group (p = 0.04). So potentially, if administering LOS in this setting in the SAH group, vasorelaxative effects will also be exerted by reactivating the vasorelaxative ET-B1-R subpart. To restore the ET-B1-R function through LOS in a non-competitive fashion, consideration might be given to the effect of SC6 under a microinflammatory environment created in the presence of AL-8810.
Additively, regarding an enhanced AT-II-1 receptor expression post-SAH 15,23, a LOS application might be beneficial by directly antagonising the AT-II-1 receptor in competitive fashion, as well. Of note, next to DCVS, DCI, DIND, and CI, poor patient outcomes after SAH is a multifactorial process depicted by neurodegeneration, early elevation of ICP, CSD and microcirculatory disturbances, and impairment of cerebral autoregulation.
Clinical context
LOS is an already well-established clinical drug commonly used for antihypertensive treatment and heart failure therapy . Therefore, it is widely available throughout the continent. Regarding all the positive effects, which have already been observed after ischemic and haemorrhagic stroke in preclinical and clinical trials 2, the question arises if there might be a room for neuroprotection even after SAH. Even more, because its application does not influence global cerebral perfusion pressure in essential hypertonic patients, which can be set equipoise to needed hypertension in patients suffering SAH. This question might be difficult to answer in a retrospective fashion because the standard of care remains that all antihypertensive agents, after the initial bleeding event, are discontinued to avoid hypotensive episodes with potential irreversible DIND during the critical phase of DCVS. Also, it is vague to postulate that the effect of LOS is long-lasting over the phase of DCVS after discontinuing it upon patient admission.”
Reviewer 2 Report
This paper reports an ingenious animal experiment, nicely designed to study a potential preventive effect of Losartan (an angiotensin-2-type-1-receptor blocker) on cerebral vasospasm elicited due to artificially evoked subarachnoid hemorrhage. The problem is generally worth of study and timely, as cerebral vessels contraction is a main source of morbidity and mortality in patients with SAH, while mechanism of this phenomenon is still not fully elucidated. Moreover, any effective treatment of this condition remains difficult, often impossible. There is also nothing bad with the set-up of this sophisticated and painstaking experiment and with methodology of processing of the obtained results. The conclusions are based on the findings.
Those are assets of this manuscript, unfortunately some shortcomings must be also pointed out. Reading the list of 30 references one can find as much as 15 which pertain to publications either of the authors of the reviewed study or to authors from their scientific institutions. Four other references came also from German centers. Certainly, it is not sin to quote yourself if there is lack of other sources, but such huge bias in citations must be spotted and the authors must assume an attitude to this problem.
It is commonly known that reading Kant is absolutely impossible for anybody who is not a kantologist and this paper shares this feature with sophisticated Kant’s philosophy. In other words - the authors showed no mercy for – say - clinical neurosurgeons and wrote the paper in a very tangled and intricate way, in particular juggling with countless acronyms to make the content just impossible to follow by somebody who is not an expert in the topic. Unexplained acronyms one can find both in the title of the contribution and in the legends of the figures and tables whereas figures and tables must be self-explaining and either described avoiding acronyms or with acronyms separately explained.
That this is ever possible (only needs much more time spent for writing) the authors have proved themselves in their earlier publications on the same topic. Let’s take an example of the contribution to Acta Neurochirurgica in 2017 ( Vascular Vasomodulatory effects of the angiotensin II type 1 receptor antagonist losartan on experimentally induced cerebral vasospasm after subarachnoid haemorrhage by Stefan Wanderer, Jan Mrosek, Gessler Florian, Seifert Volker, Juergen Konczalla). This paper has been written much more clearly and with only moderate use of acronims, because such mode of writing is usually executed by this clinically prone journal.
By the way (although it is a next important issue), both the set-up and general conclusions of this former article are vastly identical with the present one. Being a clinical neurosurgeon I am not ready to directly compare the content of both articles and find differences, also mostly because of the tangled way of presentation of the latter. That is why I would demand the authors to indicate what essentially original is presented in this new report from the old experiment in relation to the former publications of the team.
Below please find the list of references with numerous self-citations marked yellow.
References
1. Vergouwen MD, Rinkel GJ, Algra A, Fiehler J, Steinmetz H, Vajkoczy P, et al. Prospective randomized open-label trial to 445 evaluate risk factor management in patients with unruptured intracranial aneurysms: study protocol. Int J Stroke. 2018;13(9):992−8. 446
2. Wanderer S, Gruter BE, Strange F, Sivanrupan S, Di Santo S, Widmer HR, et al. The role of sartans in the treatment of stroke 447 and subarachnoid hemorrhage: A narrative review of preclinical and clinical studies. Brain Sci. 2020;10(3). 448
3. Croci DM, Wanderer S, Strange F, Gruter BE, Sivanrupan S, Andereggen L, et al. Tocilizumab reduces vasospasms, neuronal 449 cell death, and microclot formation in a rabbit model of subarachnoid hemorrhage. Transl Stroke Res. 2021. 450
4. Croci DM, Wanderer S, Strange F, Gruter BE, Casoni D, Sivanrupan S, et al. Systemic and CSF interleukin-1alpha expression 451 in a rabbit closed cranium subarachnoid hemorrhage model: An exploratory study. Brain Sci. 2019;9(10). 452
5. Asaeda M, Sakamoto M, Kurosaki M, Tabuchi S, Kamitani H, Yokota M, et al. A non-enzymatic derived arachidonyl peroxide, 453 8-iso-prostaglandin F2 alpha, in cerebrospinal fluid of patients with aneurysmal subarachnoid hemorrhage participates in the path-454 ogenesis of delayed cerebral vasospasm. Neurosci Lett. 2005;373(3):222−5. 455
6. Wanderer S, Andereggen L, Mrosek J, Kashefiolasl S, Schubert GA, Marbacher S, et al. Levosimendan as a therapeutic strategy 456 to prevent neuroinflammation after aneurysmal subarachnoid hemorrhage? J Neurointerv Surg. 2021. 457
7. Wanderer S, Gruter BE, Strange F, Boillat G, Sivanrupan S, Rey J, et al. Aspirin treatment prevents inflammation in experi-458 mental bifurcation aneurysms in New Zealand White rabbits. J Neurointerv Surg. 2022;14(2):189−95. 459
8. Frosen J, Piippo A, Paetau A, Kangasniemi M, Niemela M, Hernesniemi J, et al. Remodeling of saccular cerebral artery aneu-460 rysm wall is associated with rupture: Histological analysis of 24 unruptured and 42 ruptured cases. Stroke. 2004;35(10):2287−93. 461
9. Vatter H, Zimmermann M, Seifert V, Schilling L. Experimental approaches to evaluate endothelin-A receptor antagonists. 462 Methods Find Exp Clin Pharmacol. 2004;26(4):277−86. 463
10. Konczalla J, Vatter H, Weidauer S, Raabe A, Seifert V. Alteration of the cerebrovascular function of endothelin B receptor after 464 subarachnoidal hemorrhage in the rat. Exp Biol Med (Maywood). 2006;231(6):1064−8. 465
11. Seifert V, Loffler BM, Zimmermann M, Roux S, Stolke D. Endothelin concentrations in patients with aneurysmal subarachnoid 466 hemorrhage. Correlation with cerebral vasospasm, delayed ischemic neurological deficits, and volume of hematoma. J Neurosurg. 467 1995;82(1):55−62. 468
12. Seifert V, Stolke D, Kaever V, Dietz H. Arachidonic acid metabolism following aneurysm rupture. Evaluation of cerebrospinal 469 fluid and serum concentration of 6-keto-prostaglandin F1 alpha and thromboxane B2 in patients with subarachnoid hemorrhage. 470 Surg Neurol. 1987;27(3):243−52. 471
13. Wanderer S, Mrosek J, Gessler F, Seifert V, Konczalla J. Vasomodulatory effects of the angiotensin II type 1 receptor antagonist 472 losartan on experimentally induced cerebral vasospasm after subarachnoid haemorrhage. Acta Neurochir (Wien). 2018;160(2):277−84. 473
14. Wanderer S, Mrosek J, Vatter H, Seifert V, Konczalla J. Crosstalk between the angiotensin and endothelin system in the cere-474 brovasculature after experimental induced subarachnoid hemorrhage. Neurosurg Rev. 2018;41(2):539−48. 475
15. Konczalla J, Wanderer S, Mrosek J, Schuss P, Platz J, Guresir E, et al. Crosstalk between the angiotensin and endothelin-system 476 in the cerebrovasculature. Curr Neurovasc Res. 2013;10(4):335−45. 477
16. Kilkenny C, Browne W, Cuthill IC, Emerson M, Altman DG, Group NCRRGW. Animal research: Reporting in vivo experi-478 ments: the ARRIVE guidelines. Br J Pharmacol. 2010;160(7):1577−9. 479
17. Vatter H, Weidauer S, Konczalla J, Dettmann E, Zimmermann M, Raabe A, et al. Time course in the development of cerebral 480 vasospasm after experimental subarachnoid hemorrhage: Clinical and neuroradiological assessment of the rat double hemorrhage 481 model. Neurosurgery. 2006;58(6):1190−7; discussion 7. 482
18. Vatter H, Zimmermann M, Tesanovic V, Raabe A, Schilling L, Seifert V. Cerebrovascular characterization of clazosentan, the 483 first nonpeptide endothelin receptor antagonist clinically effective for the treatment of cerebral vasospasm. Part I: Inhibitory effect 484 on endothelin(A) receptor-mediated contraction. J Neurosurg. 2005;102(6):1101−7. 485 J. Clin. Med. 2022, 11, x FOR PEER REVIEW 14 of 14
19. Vatter H, Zimmermann M, Tesanovic V, Raabe A, Seifert V, Schilling L. Cerebrovascular characterization of clazosentan, the 486 first nonpeptide endothelin receptor antagonist shown to be clinically effective for the treatment of cerebral vasospasm. Part II: Effect 487 on endothelin(B) receptor-mediated relaxation. J Neurosurg. 2005;102(6):1108−14. 488
20. Vatter H, Konczalla J, Weidauer S, Preibisch C, Raabe A, Zimmermann M, et al. Characterization of the endothelin-B receptor 489 expression and vasomotor function during experimental cerebral vasospasm. Neurosurgery. 2007;60(6):1100−8; discussion 8−9. 490
21. Vatter H, Konczalla J, Seifert V. Endothelin related pathophysiology in cerebral vasospasm: What happens to the cerebral 491 vessels? Acta Neurochir Suppl. 2011;110(Pt 1):177−80. 492
22. Ansar S, Vikman P, Nielsen M, Edvinsson L. Cerebrovascular ETB, 5-HT1B, and AT1 receptor upregulation correlates with 493 reduction in regional CBF after subarachnoid hemorrhage. Am J Physiol Heart Circ Physiol. 2007;293(6):H3750−8. 494
23. Lobato RD, Marin J, Salaices M, Rivilla F, Burgos J. Cerebrovascular reactivity to noradrenaline and serotonin following ex-495 perimental subarachnoid hemorrhage. J Neurosurg. 1980;53(4):480−5. 496
24. Roman M, Garcia L, Morales M, Crespo MJ. The combination of dantrolene and nimodipine effectively reduces 5-HT-induced 497 vasospasms in diabetic rats. Sci Rep. 2021;11(1):9852. 498
25. Sercombe R, Dinh YR, Gomis P. Cerebrovascular inflammation following subarachnoid hemorrhage. Jpn J Pharmacol. 499 2002;88(3):227−49. 500
26. Tchekalarova JD, Ivanova NM, Pechlivanova DM, Atanasova D, Lazarov N, Kortenska L, et al. Antiepileptogenic and neuro-501 protective effects of losartan in kainate model of temporal lobe epilepsy. Pharmacol Biochem Behav. 2014;127:27−36. 502
27. Etminan N, Vergouwen MD, Ilodigwe D, Macdonald RL. Effect of pharmaceutical treatment on vasospasm, delayed cerebral 503 ischemia, and clinical outcome in patients with aneurysmal subarachnoid hemorrhage: A systematic review and meta-analysis. J 504 Cereb Blood Flow Metab. 2011;31(6):1443−51. 505
28. Tchekalarova JD, Ivanova N, Atanasova D, Pechlivanova DM, Lazarov N, Kortenska L, et al. Long-term treatment with losar-506 tan attenuates seizure activity and neuronal damage without affecting behavioral changes in a model of co-morbid hypertension and 507 epilepsy. Cell Mol Neurobiol. 2016;36(6):927−41. 508
29. Bar-Klein G, Cacheaux LP, Kamintsky L, Prager O, Weissberg I, Schoknecht K, et al. Losartan prevents acquired epilepsy via 509 TGF-beta signaling suppression. Ann Neurol. 2014;75(6):864−75. 510
30. Wanderer S, Andereggen L, Mrosek J, Kashefiolasl S, Marbacher S, Konczalla J. The role of losartan as a potential neuroregen-511 erative pharmacological agent after aneurysmal subarachnoid haemorrhage. Int J Mol Sci. 2020;21
Author Response
Reviewer 2
This paper reports an ingenious animal experiment, nicely designed to study a potential preventive effect of Losartan (an angiotensin-2-type-1-receptor blocker) on cerebral vasospasm elicited due to artificially evoked subarachnoid hemorrhage. The problem is generally worth of study and timely, as cerebral vessels contraction is a main source of morbidity and mortality in patients with SAH, while mechanism of this phenomenon is still not fully elucidated. Moreover, any effective treatment of this condition remains difficult, often impossible. There is also nothing bad with the set-up of this sophisticated and painstaking experiment and with methodology of processing of the obtained results. The conclusions are based on the findings.
REPLY: We truly thank reviewer 2 for his generous comments. We carefully took care to integrate them throughout the whole manuscript and hope the changes made are therefore for your liking.
Those are assets of this manuscript, unfortunately some shortcomings must be also pointed out. Reading the list of 30 references one can find as much as 15 which pertain to publications either of the authors of the reviewed study or to authors from their scientific institutions. Four other references came also from German centers. Certainly, it is not sin to quote yourself if there is lack of other sources, but such huge bias in citations must be spotted and the authors must assume an attitude to this problem.
REPLY: Thank you again for this valuable comment. Authors did not indeed to perform inappropriate self-citations. Therefore, we thinned out references of self-citation which do not directly deal with the endothelin-receptor- as well as losartan analyses in the double-haemorrage rat model from 30 to
- Vergouwen MD, Rinkel GJ, Algra A, et al. Prospective Randomized Open-label Trial to evaluate risk faCTor management in patients with Unruptured intracranial aneurysms: Study protocol. Int J Stroke 2018;13(9):992-998. DOI: 10.1177/1747493018790033.
- Wanderer S, Gruter BE, Strange F, et al. The Role of Sartans in the Treatment of Stroke and Subarachnoid Hemorrhage: A Narrative Review of Preclinical and Clinical Studies. Brain Sci 2020;10(3). DOI: 10.3390/brainsci10030153.
- Asaeda M, Sakamoto M, Kurosaki M, et al. A non-enzymatic derived arachidonyl peroxide, 8-iso-prostaglandin F2 alpha, in cerebrospinal fluid of patients with aneurysmal subarachnoid hemorrhage participates in the pathogenesis of delayed cerebral vasospasm. Neurosci Lett 2005;373(3):222-5. DOI: 10.1016/j.neulet.2004.10.008.
- Frosen J, Piippo A, Paetau A, et al. Remodeling of saccular cerebral artery aneurysm wall is associated with rupture: histological analysis of 24 unruptured and 42 ruptured cases. Stroke 2004;35(10):2287-93. DOI: 10.1161/01.STR.0000140636.30204.da.
- Vatter H, Zimmermann M, Seifert V, Schilling L. Experimental approaches to evaluate endothelin-A receptor antagonists. Methods Find Exp Clin Pharmacol 2004;26(4):277-86.
- Konczalla J, Vatter H, Weidauer S, Raabe A, Seifert V. Alteration of the cerebrovascular function of endothelin B receptor after subarachnoidal hemorrhage in the rat. Exp Biol Med (Maywood) 2006;231(6):1064-8.
- Seifert V, Loffler BM, Zimmermann M, Roux S, Stolke D. Endothelin concentrations in patients with aneurysmal subarachnoid hemorrhage. Correlation with cerebral vasospasm, delayed ischemic neurological deficits, and volume of hematoma. J Neurosurg 1995;82(1):55-62. DOI: 10.3171/jns.1995.82.1.0055.
- Seifert V, Stolke D, Kaever V, Dietz H. Arachidonic acid metabolism following aneurysm rupture. Evaluation of cerebrospinal fluid and serum concentration of 6-keto-prostaglandin F1 alpha and thromboxane B2 in patients with subarachnoid hemorrhage. Surg Neurol 1987;27(3):243-52. DOI: 10.1016/0090-3019(87)90037-1.
- Wanderer S, Mrosek J, Gessler F, Seifert V, Konczalla J. Vasomodulatory effects of the angiotensin II type 1 receptor antagonist losartan on experimentally induced cerebral vasospasm after subarachnoid haemorrhage. Acta Neurochir (Wien) 2018;160(2):277-284. DOI: 10.1007/s00701-017-3419-2.
- Kilkenny C, Browne W, Cuthill IC, Emerson M, Altman DG, Group NCRRGW. Animal research: reporting in vivo experiments: the ARRIVE guidelines. Br J Pharmacol 2010;160(7):1577-9. DOI: 10.1111/j.1476-5381.2010.00872.x.
- Vatter H, Weidauer S, Konczalla J, et al. Time course in the development of cerebral vasospasm after experimental subarachnoid hemorrhage: clinical and neuroradiological assessment of the rat double hemorrhage model. Neurosurgery 2006;58(6):1190-7; discussion 1190-7. DOI: 10.1227/01.NEU.0000199346.74649.66.
- Konczalla J, Wanderer S, Mrosek J, et al. Levosimendan, a new therapeutic approach to prevent delayed cerebral vasospasm after subarachnoid hemorrhage? Acta Neurochir (Wien) 2016;158(11):2075-2083. DOI: 10.1007/s00701-016-2939-5.
- Vatter H, Konczalla J, Weidauer S, et al. Characterization of the endothelin-B receptor expression and vasomotor function during experimental cerebral vasospasm. Neurosurgery 2007;60(6):1100-8; discussion 1108-9. DOI: 10.1227/01.NEU.0000255471.75752.4B.
- Vatter H, Konczalla J, Seifert V. Endothelin related pathophysiology in cerebral vasospasm: what happens to the cerebral vessels? Acta Neurochir Suppl 2011;110(Pt 1):177-80. DOI: 10.1007/978-3-7091-0353-1_31.
- Ansar S, Vikman P, Nielsen M, Edvinsson L. Cerebrovascular ETB, 5-HT1B, and AT1 receptor upregulation correlates with reduction in regional CBF after subarachnoid hemorrhage. Am J Physiol Heart Circ Physiol 2007;293(6):H3750-8. DOI: 10.1152/ajpheart.00857.2007.
- Lobato RD, Marin J, Salaices M, Rivilla F, Burgos J. Cerebrovascular reactivity to noradrenaline and serotonin following experimental subarachnoid hemorrhage. J Neurosurg 1980;53(4):480-5. DOI: 10.3171/jns.1980.53.4.0480.
- Roman M, Garcia L, Morales M, Crespo MJ. The combination of dantrolene and nimodipine effectively reduces 5-HT-induced vasospasms in diabetic rats. Sci Rep 2021;11(1):9852. DOI: 10.1038/s41598-021-89338-6.
- Sercombe R, Dinh YR, Gomis P. Cerebrovascular inflammation following subarachnoid hemorrhage. Jpn J Pharmacol 2002;88(3):227-49. DOI: 10.1254/jjp.88.227.
- Tchekalarova JD, Ivanova NM, Pechlivanova DM, et al. Antiepileptogenic and neuroprotective effects of losartan in kainate model of temporal lobe epilepsy. Pharmacol Biochem Behav 2014;127:27-36. DOI: 10.1016/j.pbb.2014.10.005.
- Etminan N, Vergouwen MD, Ilodigwe D, Macdonald RL. Effect of pharmaceutical treatment on vasospasm, delayed cerebral ischemia, and clinical outcome in patients with aneurysmal subarachnoid hemorrhage: a systematic review and meta-analysis. J Cereb Blood Flow Metab 2011;31(6):1443-51. DOI: 10.1038/jcbfm.2011.7.
- Tchekalarova JD, Ivanova N, Atanasova D, et al. Long-Term Treatment with Losartan Attenuates Seizure Activity and Neuronal Damage Without Affecting Behavioral Changes in a Model of Co-morbid Hypertension and Epilepsy. Cell Mol Neurobiol 2016;36(6):927-941. DOI: 10.1007/s10571-015-0278-3.
- Bar-Klein G, Cacheaux LP, Kamintsky L, et al. Losartan prevents acquired epilepsy via TGF-beta signaling suppression. Ann Neurol 2014;75(6):864-75. DOI: 10.1002/ana.24147.
- Vikman P, Ansar S, Henriksson M, Stenman E, Edvinsson L. Cerebral ischemia induces transcription of inflammatory and extracellular-matrix-related genes in rat cerebral arteries. Exp Brain Res 2007;183(4):499-510. DOI: 10.1007/s00221-007-1062-5.
Main references of our group, to draw the red line from the beginning of the endothelin-receptor characterization until testing losartan regarding its function as neuroprotective drug had to be included in this manuscript.
It is commonly known that reading Kant is absolutely impossible for anybody who is not a kantologist and this paper shares this feature with sophisticated Kant’s philosophy. In other words - the authors showed no mercy for – say - clinical neurosurgeons and wrote the paper in a very tangled and intricate way, in particular juggling with countless acronyms to make the content just impossible to follow by somebody who is not an expert in the topic.
REPLY: Thank you for your commentary. We simplified the paper in terms of readability and removed most of the acronyms, except SAH, 5-HT, DCVS, CEC, Emax, mM, mg, kg, LOS, AT-II-1, AT-II-2, ET-1, ET-A-R and ET-B1-R.
Unexplained acronyms one can find both in the title of the contribution and in the legends of the figures and tables whereas figures and tables must be self-explaining and either described avoiding acronyms or with acronyms separately explained.
REPLY: Thank you again, we removed unexplained acronyms in the title, in the legends of figures and tables throughout the whole manuscript.
That this is ever possible (only needs much more time spent for writing) the authors have proved themselves in their earlier publications on the same topic. Let’s take an example of the contribution to Acta Neurochirurgica in 2017 ( Vascular Vasomodulatory effects of the angiotensin II type 1 receptor antagonist losartan on experimentally induced cerebral vasospasm after subarachnoid haemorrhage by Stefan Wanderer, Jan Mrosek, Gessler Florian, Seifert Volker,Juergen Konczalla). This paper has been written much more clearly and with only moderate use of acronims, because such mode of writing is usually executed by this clinically prone journal.
REPLY: We hope that by removing most of the acronyms the paper has gained more readability and is therefore for your liking.
By the way (although it is a next important issue), both the set-up and general conclusions of this former article are vastly identical with the present one. Being a clinical neurosurgeon I am not ready to directly compare the content of both articles and find differences, also mostly because of the tangled way of presentation of the latter.
REPLY: Thank you again for your valuable comments. With the article entitled “Vasomodulatory effects of the angiotensin II type 1 receptor antagonist losartan on experimentally induced cerebral vasospasm after subarachnoid haemorrhage”, authors analyzed the effect of losartan on prostaglandin F2a-enhanced vasoconstriction after SAH and in sham-operated rats. Authors pointed out that losartan reduced PGF2a-induced vasoconstriction and reversed a PGF2a-precontraction completely. In the actual experimental series, authors specifically analyzed the effect of losartan on vasocontraction induced by ET-1 and 5-HT, other peptides being enhanced synthetized after SAH and responsible for the development of DCVS. The experimental setting by the use of a double-haemorrhage-rat model stayed the same.
That is why I would demand the authors to indicate what essentially original is presented in this new report from the old experiment in relation to the former publications of the team.
REPLY: Thank you again for your precious comment. We tried to reformulate the first paragraph of the discussion clearer from
"Our findings demonstrate that LOS possesses the ability to diminish experimentally induced DCVS in a proinflammatory microenvironment created by the application of 5-HT and ET-1 in sham segments. Besides a concentration-dependent antagonisation of an already known ET-1-induced vasocontraction, a 5-HT-mediated vasoconstriction of basilar artey ring segments was antagonised as well in a dose-dependent manner 9, even stronger with the application of LOS 10-3 M."
to
"To the best of authors' knowledge our experimental findings demonstrate for the first time that LOS applied in a concentration of 10-3 M possesses the ability to diminish experimentally induced DCVS in a proinflammatory microenvironment, created by the application of 5-HT and ET-1 –both key players in the development of cerebral vasospasm– in sham segments. Besides a concentration-dependent antagonisation of an already known ET-1-induced vasocontraction, a 5-HT-mediated vasoconstriction of basilar artey ring segments was antagonised in a dose-dependent manner 9, as well. "
Round 2
Reviewer 2 Report
The authors have made modifications of their manuscript in a line with the submitted remarks. Although still not too profound, with them the paper seems acceptable for publication.